# Production, Characterization, and In Vitro and In Vivo Studies of Nanoemulsions Containing St. John’s Wort Plant Constituents and Their Potential for the Treatment of Depression

**DOI:** 10.3390/ph16040490

**Published:** 2023-03-26

**Authors:** Ahmad Salawi, Yosif Almoshari, Muhammad H. Sultan, Osama A. Madkhali, Mohammed Ali Bakkari, Meshal Alshamrani, Awaji Y. Safhi, Fahad Y. Sabei, Turki Al Hagbani, Md Sajid Ali, Md Sarfaraz Alam

**Affiliations:** 1Department of Pharmaceutics, College of Pharmacy, Jazan University, Jazan 45142, Saudi Arabia; yalmoshari@jazanu.edu.sa (Y.A.); mhsultan@jazanu.edu.sa (M.H.S.); omadkhali@jazanu.edu.sa (O.A.M.); mbakkari@jazanu.edu.sa (M.A.B.); malshamrani@jazanu.edu.sa (M.A.); asafhi@jazanu.edu.sa (A.Y.S.); fsabei@jazanu.edu.sa (F.Y.S.); msali@jazanu.edu.sa (M.S.A.); msalam@jazanu.edu.sa (M.S.A.); 2Pharmaceutics Department, College of Pharmacy, University of Hail, Hail 55473, Saudi Arabia; t.alhagbani@uoh.edu.sa

**Keywords:** hypericin, chitosan, nanoemulsion, stability studies, depression

## Abstract

The current project was designed to prepare an oil-in-water (oil/water) hypericin nanoemulsion using eucalyptus oil for the preparation of an oil phase with chitosan as an emulsion stabilizer. The study might be a novelty in the field of pharmaceutical sciences, especially in the area of formulation development. Tween^®^ 80 (Polysorbate) was used as the nonionic surfactant. The nanoemulsion was prepared by using the homogenization technique, followed by its physicochemical evaluation. The surface morphological studies showed the globular structure has a nano-sized diameter, as confirmed by zeta size analysis. The zeta potential analysis confirmed a positive surface charge that might be caused by the presence of chitosan in the formulation. The pH was in the range of 5.14 to 6.11, which could also be compatible with the range of nasal pH. The viscosity of the formulations was found to be affected by the concentration of chitosan (F1-11.61 to F4-49.28). The drug release studies showed that the presence of chitosan greatly influenced the drug release, as it was noticed that formulations having an elevated concentration of chitosan release lesser amounts of the drug. The persistent stress in the mouse model caused a variety of depressive- and anxiety-like behaviors that can be counteracted by chemicals isolated from plants, such as sulforaphane and tea polyphenols. In the behavioral test and source performance test, hypericin exhibited antidepressant-like effects. The results show that the mice treated for chronic mild stress had a considerably higher preference for sucrose after receiving continuous hypericin for 4 days (*p* = 0.0001) compared to the animals administered with normal saline (*p* ≤ 0.0001) as well as the naïve group (*p* ≤ 0.0001). In conclusion, prepared formulations were found to be stable and can be used as a potential candidate for the treatment of depression.

## 1. Introduction

Nanoemulsions are formulations with nano-sized globular structures that are currently being extensively researched as medication carriers to enhance the delivery of therapeutic agents. The droplet sizes of nanoemulsions typically fall in the range of 20–200 nm as well as showing narrow size distributions. These emulsions show promising results for upcoming products for use in drug therapies, diagnosis, and cosmetics, as well as in biotechnologies [1]. These nanoemulsions, which are stabilized by the addition of a suitable surfactant, can be either water-in-oil (water/oil) or oil-in-water (oil/water) dispersions of the two immiscible liquids [2]. Although it should not be mistaken with microemulsions, the term “nanoemulsion” is occasionally used synonymously with “mini-emulsion” or “submicron emulsion” [3,4]. In addition to being employed in a variety of dosage forms, including liquids, aerosols, gels, sprays, and creams, nanoemulsions can also be administered by a variety of different routes, including topical, intranasal, ophthalmic, intravenous, and oral [5,6]. The release of drugs from the nanoemulsions includes their breakdown from an oil into the surfactant layer as well as into the aqueous phase. While diffusing out of the oil, the drug’s solubilized moiety meets nearby water and undergoes nanoprecipitation. According to the Noyes–Whitney equation, this greatly increases the surface area of the drugs and thus its dissolution. To achieve a controlled or sustained release formulation, the dynamics of drug release could be changed at each of these steps by slightly altering the composition of the nanoemulsion [7].

Nanoemulsions could be utilized efficiently to disguise the metallic or bitter taste of drugs, which causes nausea as well as being associated with non-compliance to the patients. These types of formulations could be employed to improve the bioavailability of drugs with poor water solubility by formulating oil/water (o/w) nanoemulsions. Nanoemulsions commonly contain 5–20% droplets of oil or lipids in the case of oil/water emulsions. This proportion can be increased significantly (up to 70%), occasionally or as needed [8]. Surfactants are basically amphiphilic molecules that are used to stabilize nanoemulsions by decreasing the interfacial tension and preventing droplet aggregation. They tend to be adsorbed quickly at the oil–water interface and then provide electrostatic or steric or dual electro-steric stabilization. In nanoemulsions, the commonly used surfactants are Tween 20, 40, 60, and 80 (Polysorbate) [9,10] and Span 20, 40, 60, and 80 (Sorbitan monolaurate) [11,12].

The common plant St. John’s Wort (Hypericum species) contains the naturally occurring compound hypericin, which can also be produced via the anthraquinone-derived emodin. Hypericin is a plant-derived substance, which in the past has been used as medicine [13,14]. The first reported study on the isolation of hypericin from *Hypericum perforatum* (*H. perforatum*) was published in 1939 [15]. Brockmann et al. published the first accurate chemical formula (C_30_H_16_O_8_) of hypericin and the correct chemical structure in 1942 and 1950, respectively [16,17]. Hypericin is hydrophobic as well as insoluble in methylene chloride and in various other solvents that are nonpolar. In recent years, hypericin has been defined as an agent with antiviral activity against human immunodeficiency virus and other viruses [18,19]. At present, it is still of notable interest and is also one of the major topics of discussion regarding its potential scope for clinical applications. There are different therapeutic applications of hypericin, including anti-neoplastic, anti-viral, photodynamic, anti-retroviral, and anti-tumor as well as photo-diagnostic activities, which are presently under study. Furthermore, currently, hypericum extract is still widely used for treating mild as well as moderate depression [20]. Although many synthetic antidepressants are widely prescribed in clinics, they carry a high risk of relapse as well as numerous side effects [21]. Antidepressants that are derived from natural and herbal sources are expected to have fewer negative effects [22,23]. Many clinical studies have shown that St. John’s Wort is just as effective as the commonly used synthetic antidepressants. St John’s Wort (*Hypericum perforatum*), a flowering plant, has been used for centuries in the United States and Europe to treat mild to moderate depression [23]. Since 1984, Commission E in Germany has licensed it for use in the treatment of anxiety, depression, and insomnia [24].

The current paper demonstrates the preparation of a stable oil/water nanoemulsion of hypericin, followed by its physicochemical, in vitro, and in vivo evaluations. The prepared formulation was also evaluated for its anti-depressant activities.

## 2. Results and Discussion

The lambda max was found to be 243 nm as described in Appendix A. Although several noticeable peaks were observed, a good linearity was found at the selected wavelength. The calibration curve was drawn, showing a good linearity behavior at the selected wavelength, confirming the suitability of the selected lambda max. The coefficient of correlation was closer to one, i.e., 0.9988.

### 2.1. Characterization of the System

The particle size analysis was conducted to confirm the nano-dispersions (Table 1). Globules were in the nano range, ranging from 130 ± 06 (F1) to 141 ± 05 (F4). The size was comparable to that previously reported by Ishkeh et al. (2021) [25]. There was no notable change in the size range, and the slight differences could be caused by the increase in the viscosity of the formulations that had a comparatively greater concentration of chitosan [26].

It can be noticed that the globules were of uniform size distribution and well segregated from each other, which was possibly due to the viscosity because of the presence of chitosan, which kept the globules apart and prevented aggregation. Another fact might be the positive charge of the globules, due to the presence of cationic polymer (chitosan). The zeta potential analysis showed the potential was +56.5 (F3) to +89.6 (F4) mV, which also forced the globules to be segregated, hence maintaining their nanosized range.

The findings disclosed a considerable surface charge of presented in Table 1 and Appendix A. In current formulations, chitosan is used as a stabilizing and thickening agent. It also provides suitable mucoadhesion. It belongs to the cationic group of polymers, bearing a positive charge. This might be the cause for the globules having a suitable charge on the surface, causing globules to be in a well-segregated form. The chitosan, being a cationic polymer, can add a positive charge on the surface, and these findings were also observed in previous studies [27].

### 2.2. Morphology Analysis of the System

The morphology of the developed system was analyzed using scanning electron microscopy. The outcomes revealed that a nanosized, slightly spherically shaped globule was detected. Furthermore, well-dispersed and segregated globules were noticed, indicating a stable formulation. The SEM analysis is in accordance with the zeta potential analysis, which showed that the globules have a reasonable surface charge, causing them to remain separate (Figure 1 and Appendix A).

### 2.3. Dilutability Test

The nanoemulsion has been prepared successfully (Appendix A), and evaluated for different parameters, and one of the important was dilutability test. The dilution test was developed to demonstrate that a nanoemulsion’s stability can be increased by adding a continuous phase in larger amounts [28]. When water was added for the dilution of the formulation, it diluted easily, confirming the aqueous medium as an external phase and, ultimately, as an oil/water emulsion. Furthermore, the phase separation was not observed (Figure 2).

### 2.4. Dye Solubility Test

In an oil/water globule, a hydrophilic color is dispersed, but is soluble in the aqueous phase. On the other hand, an oil-soluble dye is soluble in the oily phase of the oil/water globule, but dispersible in the water/oil globule [29]. The addition of amaranth dye was used to mark the continuous aqueous phase, but not the dispersed oily phase at 40× (Figure 3). According to the test’s findings, the nanoemulsion was confirmed as an oil-in-water emulsion.

### 2.5. pH, Refractive Index, and Viscosity

The pH of the nanoemulsion was found to be in the range of 5.14–6.11, which might be compatible with the physiological pH of different body cavities, for example, the oral and nasal cavities. Hence, it can be non-irritating to patients. The refractive index was in the range of 1.3422 to 1.3491, while the viscosity of the prepared nanoemulsion was found to increase with increase in the concentration of chitosan. Chitosan is a well-known viscosity enhancer and emulsion stabilizer, which was observed in the current study [30,31] (Table 2).

### 2.6. In Vitro Drug Release Studies

The drug release from the cellophane membrane was studied by drawing samples after 5, 10, 20, 30, and 60 min and evaluating them spectrophotometrically at 243 nm wavelengths, and the absorbance of each sample was recorded. It was observed that all the formulations showed different patterns of drug release at different time intervals. The behavior and pattern of the drug release from most of the formulation were found to be affected by the varying concentrations of chitosan in each formulation. However, at the end of the study, where it was expected that chitosan and other ingredients were dissolved, most of the drug was released from the prepared formulations (Figure 4).

According to the literature, the solubility of a drug can be improved up to 0.25 mg/mL by mixing with the organic phase, followed by the addition of buffer or water. In the current study, the same approach was employed to prepare the nanoemulsion. An alcoholic solution of the drug was prepared, followed by mixing with other constituents of the organic phase. The ultimate concentration of the drug in each formulation was 2.5 mg/10 mL (0.25 mg/mL). The drug diffused across the membrane slowly, over the course of 60 min, hence converting it to a more diluted form. This might be the probable reason that most of the drug was released from the formulations. The literature also suggested that, as chitosan is a mucoadhesive polymer, it can improve the permeability of drugs both in vitro and in vivo [32,33]. It could also be predicted that, due to the positive zeta potential, the globules will become attached with the negatively charged mucosal membrane, providing an opportunity for the drug to permeate easily and effectively. Hence, the addition of chitosan in the formulation could add the benefits of enhanced permeability, leading to the improved bioavailability of the drug. This effect could be evaluated in the future by permeation studies, using different ex vivo or in vivo models. Furthermore, it has a great influence on the release of drugs from formulations, as different researchers have reported its drug retardation effect.

### 2.7. Drug Solubility Studies

The solubility of the drug in the buffer medium was observed to be 4.3 ± 0.103 µg/mL, which confirmed the poor water solubility of the drug.

### 2.8. Stability Studies

After centrifugation, the formulations were observed regarding their stability. It was observed that there were no signs of phase separation, cracking, or creaming. The formulations were also subjected to the freeze–thaw method and found to be stable. Stoke’s law states that centrifugation affects emulsions by destabilizing the distributed fat, which causes the level of fat globules to increase [34,35]. Recent studies have suggested that, in both industrial and scientific contexts, the process of centrifugation involves using a centrifuge to use the centrifugal force to separate heterogeneous mixtures. However, after performing the test, no sedimentation of the layers or particles was observed.

Similarly, the stability was also confirmed by the outcomes of the freeze–thaw method. The results confirm that there is no sedimentation or oil separation. Hence, the stability of the formulations was at a satisfactory level (Figure 5).

### 2.9. In Vivo Studies

Based on the previous tests performed, F1 Nano-emulsion had been selected for the in-vivo studies, as it had most prominently showed a better drug release profile; initially, the release rate was slow and the extent was greater at the end of the studies. Similarly, the particle size was smallest amongst all of the four formulations, etc. Hence, keeping in mind the satisfactory outcome, F1 was carried out for further analysis.

#### 2.9.1. Forced Swim Test (FST)

Antidepressant drugs are commonly screened using the forced swim test [36]. Tricyclics, MAO inhibitors, 5-HT-specific reuptake inhibitors, and atypicals are only a few of the major antidepressant medication families, for which this test is extremely sensitive and specific [37]. In FST, mice are made to swim against their will in a small area from which they cannot escape, and they are made to display a typical immobility behavior. Numerous drugs that are therapeutically effective in human depression reduce this behavior, which reflects a state of despair. The tail suspension test, such as in the forced swim test, induces a state of despair in animals. This immobility, which is referred to as “behavioral despair” in animals, is thought to imitate a condition such as sadness in humans [38].

In our study, animals administered with hypericin demonstrated an improved performance compared to the normal saline group (Figure 6; *p* ≤ 0.0001). To our surprise, the acute antidepressant effect of hypericin surpassed that of the standard despramine drug (*p* ≤ 0.0001). The significant difference (*p* ≤ 0.0001) in the average immobility times (s) between the NS and HYP groups is suggestive of the FST’s ability to induce a depression-like state in mice, which is well preserved by hypericin administration in the test.

#### 2.9.2. Tail Suspension Test

Similarly, the tail suspension test is a well-practiced screening test for evaluating antidepressant drugs [39]. The test is simple but effective in comparing the antidepressant potentials of investigative drug with the conventionally available standard drugs. An incredibly interesting result of the present study is demonstrated in Figure 7, where hypericin administration improved the mobility under duress in the tail suspension test compared to the NS group (*p* ≤ 0.0001) and DES group (*p* ≤ 0.0001). Considering the fact that false-positive results can pollute the authenticity of the results, we performed the tail suspension test twice in the treated animals and compared the results of both these groups, which were non-significant (*p* = 0.3721; not shown here). The question of how hypericin reduces the depressive impact of the hanging exercise is not known. However, the literature highlights that pharmacologically active substances demonstrate their antidepressant effects by stimulating locomotor activity [40,41]. This might suggest that hypericin increases the mobility time in mice subjected to the tail suspension test. Tricyclic antidepressants (TCAs) such as desipramine specifically prevent norepinephrine (noradrenaline) from reabsorbing from the neural synapse. With hypericin administration, the increased mobility in the HYP group (*p* ≤ 0.0001), compared to DES-administered animals, shows that hypericin is involved in multiple neuronal pathways to improve depression-like symptoms.

#### 2.9.3. Sucrose Preference Test

Numerous studies have shown that stress, especially persistent stress, plays a significant role in the development of depression [42]. Persistent stress in mouse models may cause a variety of depressive- and anxiety-like behaviors, such as decreased sucrose preference in the source performance test, which can be counteracted by chemicals isolated from plants, such as sulforaphane and tea polyphenol [43]. A behavioral test, source performance test, and a chronic mild stress-induced depression mouse model were used to determine if hypericin also exhibits antidepressant-like effects. The results shown in Figure 8A reveal that mice treated for chronic mild stress had a considerably higher preference for sucrose after receiving continuous hypericin for 4 days (*p* = 0.0001), compared to the animals administered with the vehicle (NS; *p* ≤ 0.0001) as well as the naïve group (*p* ≤ 0.0001). Surprisingly, hypericin significantly increased the sucrose preference (%) in mice compared to the standard drug, i.e., desipramine (*p* ≤ 0.0001), suggesting that hypericin might be involved in diverse anti-depressant mechanisms. Previous research on this model has shown that chronic stress/persistent stress has been linked to increased oxidative stress, inflammation, and depression [44,45]. The development of antidepressants is increasingly utilizing for treatments that are anti-oxidative and anti-inflammatory [43]. Several molecules, including nuclear factor kappa B (NF-B), cyclooxygenases (COX1/COX2), inducible nitric oxide (NO) synthase (iNOS), and NO, have been connected to the onset of depression and the use of antidepressants [43,46]. Hypericin’s antioxidant, anti-inflammatory, and other pharmacological activities have been reported in the literature [47]. We could rely on the previously mentioned pharmacological activators of hypericin in reducing depression-like symptoms in mice, such as anti-inflammatory effect by inhibiting NO generation, reducing iNOS activity, and increasing the expression of heme oxygenase 1 (HO-1) [48]. This increased preference in sucrose was noted without any effect in the food intake (g) of animals in any of the groups (Figure 8B; *p* = 0.9314).

## 3. Materials and Methods

### 3.1. Materials

Hypericin, low molecular weight chitosan (stabilizer and viscosity enhancer) were purchased from Sigma Aldrich chemie Gmbh USA (St. Louis, MI, USA), eucalyptus oil (The major constituents of eucalyptus leaves essential oils are 1,8-cineol ranging from 49.07 to 83.59%, and α-pinene in the range of 1.27 to 26.35%), ethanol, Tween^®^ 80, Span^®^ 80, and distilled water were purchased from Dawaa Al-Gharbyah, KSA. All of the chemicals used were of the highest analytical grade.

### 3.2. Method

The nanoemulsion (Appendix A) was prepared by using a high shear mixer, i.e., a homogenizer. Four formulations were prepared by using variable concentrations of chitosan (Table 3). A 1% alcoholic solution of hypericin was prepared in a beaker by using magnetic stirrer for 15–20 min. In order to prepare the oil phase, 1 mL of eucalyptus oil along with 0.5 mL of Span^®^ 80 were mixed in a beaker magnetic stirrer, followed by dropwise addition of 1.5 mL of hypericin solution, under continuous stirring for 30 min. For the preparation of the aqueous solution, initially, chitosan was soaked and stirred overnight in 1% acetic acid solution. The aqueous phase was prepared by adding 0.5 mL of Tween^®^ 80, followed by the addition of the chitosan solution under continuous stirring, until a homogeneous solution was formed (Table 3). Subsequently, the oil phase was slowly added to the aqueous phase while continuously stirring to form a mixture, which was then homogenized for 30 min at 15,000 rpm using homogenizer (SilentCrusher M (EU), Heidolph Instruments Gmbh & Co. KG, Schwabach, Germany).

### 3.3. Characterization of Nanoemulsions

The lambda max was determined by using double-beam UV/Visible-Spectrophotometer (LC 95, Perkin Elmer, Waltham, MA, USA). The solution of hypericin 10 µg/mL was prepared and scanned in the range of 190 to 800 nm.

The calibration curve was constructed by preparing a 1 mg/mL solution of drug in ethanol, followed by the preparation of serial dilution by ethanol ranging from 2.5 to 20 µg/mL. The absorbance of each dilution was observed by UV/Visible Spectrophotometer (LC 95, Perkin Elmer, Waltham, MA, USA) at 243 nm. Ethanol was used as reference to exclude its impact from the absorbance at 243 nm. A curve was drawn between the concentration on the *x*-axis and the relative absorbance on the *y*-axis.

An ideal pH is needed for the stability of the emulsion. The pH was determined by the use of ADWA pH meter (AD1050) at room temperature (25 °C), and the readings were recorded.

The viscosity values of the formulations were determined by using a viscometer (Brookfield-model RV-III, New Castle, DE, USA) at 50 rpm using spindle no. 2 and the readings were recorded at 25 °C.

The refractive index is described as the relationship between the phase speed (vp) of the wave in the medium and the speed (c) of the wave in a reference medium, such as light or sound. It is expressed as:n = c/vp

The refractive index of the samples was calculated by the use of refractometer (Bellingham and Stanley, Tunbridge Wells, UK), and the readings were recorded.

#### 3.3.1. Zeta Potential

The zeta potential was evaluated by using an instrument known as the ZetaPALS (Brookhaven Instruments Corporation, Holtsville, NY, USA). Measurements of the zeta potential were carried out by the dilution of the nanoemulsion formulations and then its values were determined by the electrophoretic mobility of the droplets of oil at room temperature [49].

#### 3.3.2. Dilutability Test

When diluted with water, oil/water nanoemulsions are dilutable, whereas water/oil nanoemulsions are not. This was tested by diluting 1 mL of the optimized nanoemulsion with 10 mL of water and checking the results in a test tube.

#### 3.3.3. Dye Solubility Test

A hydrophilic dye is dispersible in the oil/water globule but solubilized in the aqueous phase of the water/oil globule. To test this, 1 mL of nanoemulsion in an Eppendorf tube was thoroughly mixed with a few drops of the hydrophilic color (amaranth). Under a fluorescent inverted microscope (DFC310 FX, Leica, Solms, Germany), the formulation was examined.

#### 3.3.4. Scanning Electron Microscopy (SEM)

A scanning electron microscope (EVO LS10 Zeiss Germany) with point-to-point resolution was used to study the morphology and structure of the nanoemulsion droplets [50]. A drop of the nanoemulsion was immediately applied to the grid of the holey film to perform the SEM observations, and the images were obtained after drying [51].

#### 3.3.5. In Vitro Drug Release Studies

Drug release studies were performed by using the USP Rotating Paddle Dissolution Apparatus (USP II Sotax Smart AT7 Dissolution Tester) at 37 °C and 50 rpm. Vessels of the apparatus were filled with phosphate buffer solution (pH 6.8), and 1 mL of each formulation was poured on the dialysis membrane while the dialysis membrane was tied up with the help of thread from one end. After pouring 1 mL of each formulation on each of the dialysis membranes, the membranes were tied from the other end as well, then were hung by thread to the paddle of each vessel already containing the buffer solution. To maintain the sink conditions, 500 mL of the dissolution medium was maintained at a constant level. The samples, each of 1 mL, were taken at different intervals of time and suitably diluted, followed by UV analysis to quantify the released drug [52].

#### 3.3.6. Drug Solubility Studies

Drug solubility studies were performed with the same medium used in drug release studies (phosphate buffer solution). A total of 1 mg of the drug was taken and added in a sufficient volume of the buffer to make the concentration 1 mg/mL. The mixture was vortexed for 5 min, followed by sonication for 20 min to ensure the maximum possible solubility of the drug in the buffer. The resultant solution was centrifuged for 5 min at 6000 rpm. The centrifugated solution was then filtered with 0.22 µm filter paper, and the absorption was checked using a UV spectrophotometer, followed by the determination of the extent of solubility.

#### 3.3.7. Stability Studies

Stability studies of these prepared formulations were performed. Firstly, the nanoemulsions were passed through the centrifugation studies in which the formulated nanoemulsions were centrifuged (Sorvall Legend RT centrifuge, US) at 5000 rpm for 30 min and then examined for any phase separation, cracking, or creaming [53].

The emulsions, after being passed through centrifugation and stability studies, were determined by the freeze–thaw method, in which the nanoemulsion formulations were transferred to NMR tubes for freeze–thaw cycling. Then, the samples were frozen daily in a −80 °C freezer for 17 ± 2 h and then thawed at 40 °C for 1 h and, at last, stored at 25 °C for analysis [53]. In general, the water removal ratio of the oil/water emulsion during the freeze/thaw process is determined by the initial water content, freezing temperature, freezing time, thawing rate, and thawing temperature. The better the dewatering results, the gentler the thawing process. The best thawing conditions are either in ambient air or in a water bath at temperatures below 20 °C [54,55].

#### 3.3.8. In Vivo Studies

##### Forced Swim Test (FST)

The method used in this paper was described by Porsolt et al. (1977). Animal studies were conducted in accordance with the recommendations of the Institutional Animal Ethical Committee for Wa’ed project (reference number W41-041). The Institutional Animal Ethical Committee, College of Pharmacy, Jazan University, Jazan, KSA, authorized the experimental protocols. In this study, the animals were divided into three groups of ten at random. Orally, Group I (control) received 0.9% saline water (10 mL/kg). Group II was administered desipramine 20 mg/kg (i.p.). Group III was provided 1.3 mg/kg (p.o.) hypercin. Individual mice were made to swim in a 15 cm deep, open cylindrical container with a diameter of 10 cm and a height of 25 cm. The test lasted for six minutes, and the total amount of time the mice were immobile was evaluated as reported by Zomkowski et al. (2005). Every mouse was judged to be immobile when it gave up and floated motionless in the water, only moving enough to maintain its head above the surface. The period of inactivity was timed. Antidepressant activity was evaluated using the FST’s duration of immobility [37].

##### Tail Suspension Test

Steru et al.’s study used computerized tail suspension test. BSTST2CA with BSTST2LOG software was utilized to measure the overall duration of immobility brought on by tail suspension (1985). In brief, the isolated mice in each chamber were suspended by adhesive tape 1 cm from the tip of the tail. The total duration of inactivity was determined during the last 4 min of a 6-min test [56]. For the tail suspension test, three groups of mice were assigned at random; Group I (control) were administered 0.9% saline water (10 mL/kg) orally. Desipramine 20 mg/kg (i.p.) was administered to Group II. Group III was administered 1.3 mg/kg (p.o.) hypericin.

#### 3.3.9. Chronic Mild Stress Mouse Model and Sucrose Preference Test

Based on previous research [43], a chronic mild stress model of depression in mice was developed. For 28 days, mice were subjected to two randomly chosen stressors: 2 h of restraint, 5 min of forced cold swimming, 24 h of water deprivation, 12 h of food deprivation, tilted cages, soiled bedding, 24 h of light/dark cycle reversal, 1 min of tail clenching, 24 h of tight environment, and 24 h of bedding deprivation. The chronic mild stress mice were randomly divided into four sub-groups beginning on the 21st day of their 28-day treatment: Group I (Naive group), Group II (Normal Saline group), Group III (Desipramine group/standard treatment group), and Group IV (Hypericin/treatment group). Mice in the Naive group were kept stress-free in their home cage for 28 days. Behavioral tests were carried out 24 h after the last administration. A source performance test was carried out in accordance with the previous description [43]. Groups II, III, and IV were administered normal saline, desipramine, and hypericin once daily from day 24 to day 28 of the chronic mild stress. Mice were acclimatized for 24 h to two bottles of a 1% sucrose solution on the 28th day, followed by 24 h of water deprivation prior to the source performance test. After that, the mice were placed in individual cages and afforded free access to two bottles, one containing water and the other containing a 1% sucrose solution, for 24 h. The amount of water and sucrose consumed for 24 h were measured at the end of source performance test. The preference for sucrose was quantified as a ratio between the amount of sucrose solution consumed and the total amount of liquid consumed [43].

## 4. Conclusions

A stable and suitable nanoemulsion formulation having globular size of less than 150 nm was developed, and extensive characterization was performed. The addition of chitosan imparted an appropriate surface charge of 58 mV to the globule, making it more stable and segregated. The in vivo studies proved that hypericin can be a potential candidate for the treatment of depression. In the future, similar studies can be performed using different polymers and the behaviors of nanoemulsions can be checked. Different types of formulations can also be created and comparison studies must be performed to assess which formulation has good and effective results.

## Figures and Tables

**Figure 1 pharmaceuticals-16-00490-f001:**
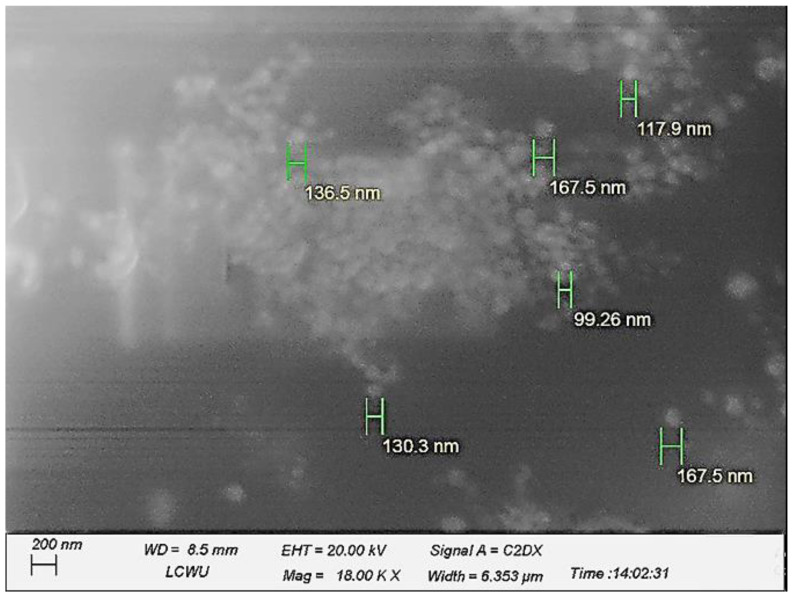
Scanning electron microscopy (F1 formulation), indicating the nano-sized globules with an appropriate spherical shape, well apart from each other.

**Figure 2 pharmaceuticals-16-00490-f002:**
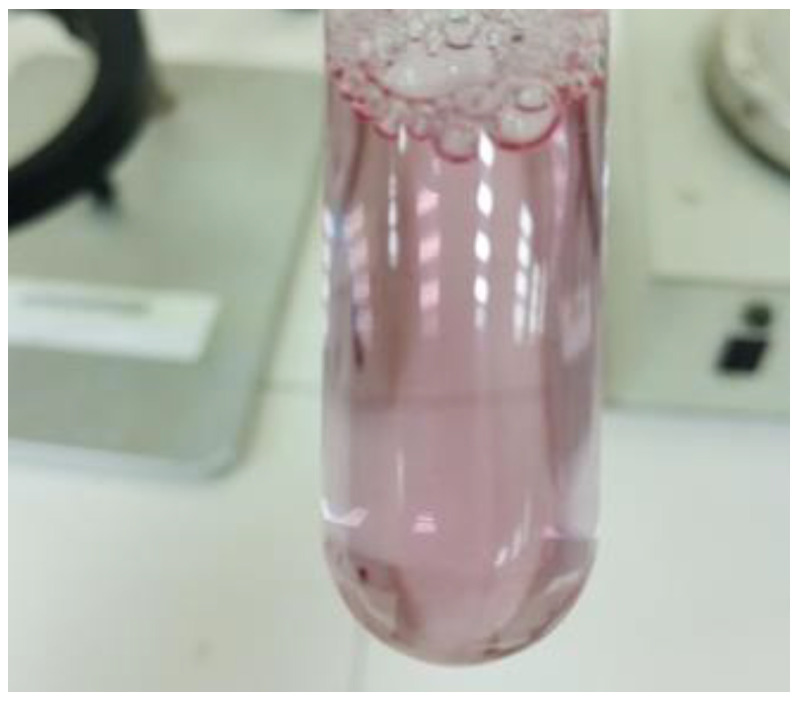
Dilution test (F1 formulation), indicating no phase separation after diluting with water.

**Figure 3 pharmaceuticals-16-00490-f003:**
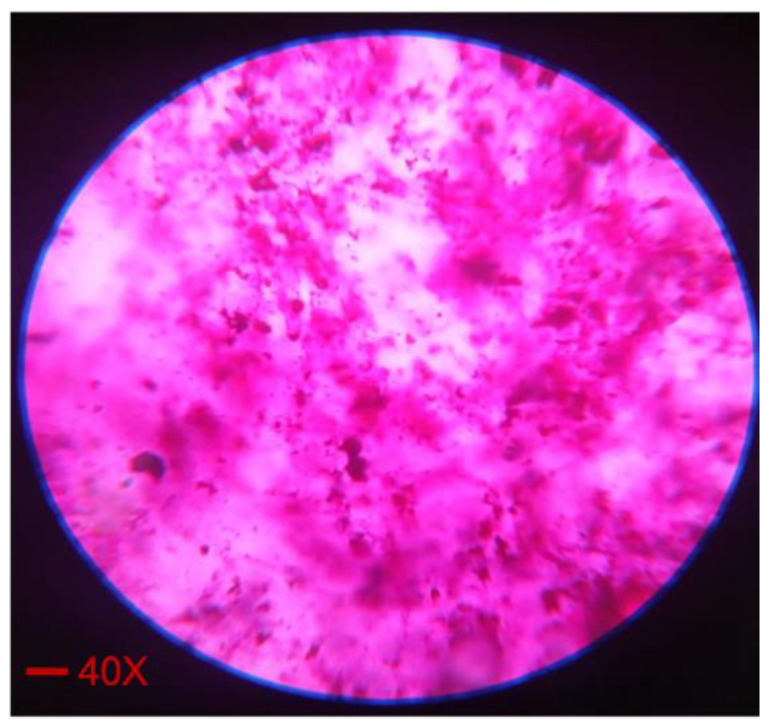
Microscopic image (F1 formulation) indicating the solubility of amaranth in the continuous phase at 40×.

**Figure 4 pharmaceuticals-16-00490-f004:**
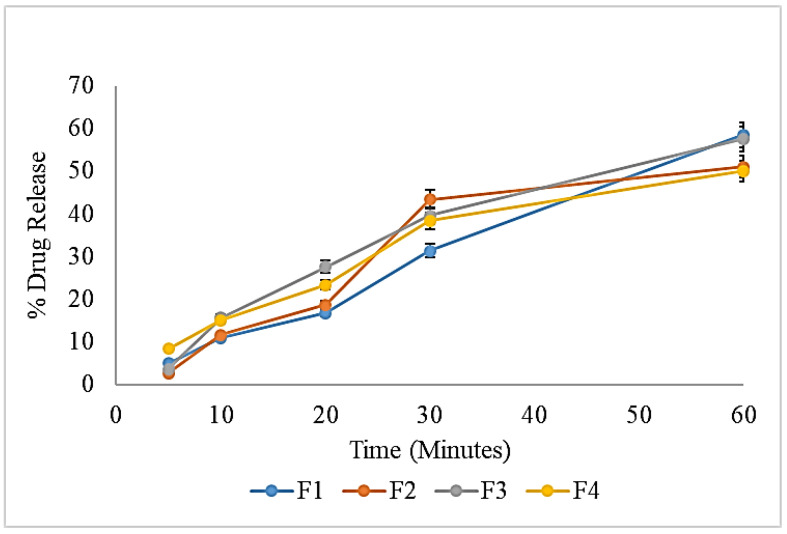
Graphical representation of the drug release studies from the prepared nanoemulsion.

**Figure 5 pharmaceuticals-16-00490-f005:**
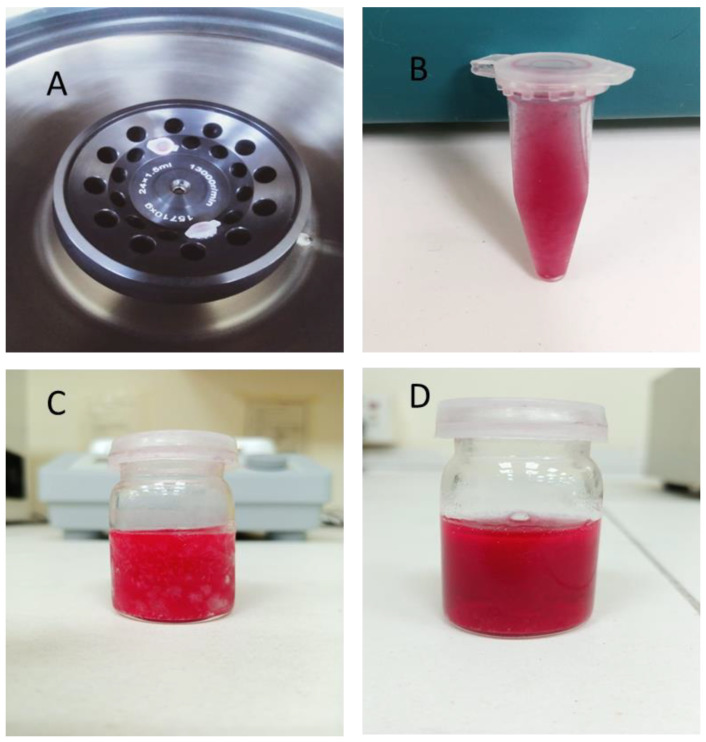
Stability of the F1 formulation both after centrifugation (**A**,**B**) and with freeze–thaw (**C**,**D**) methods.

**Figure 6 pharmaceuticals-16-00490-f006:**
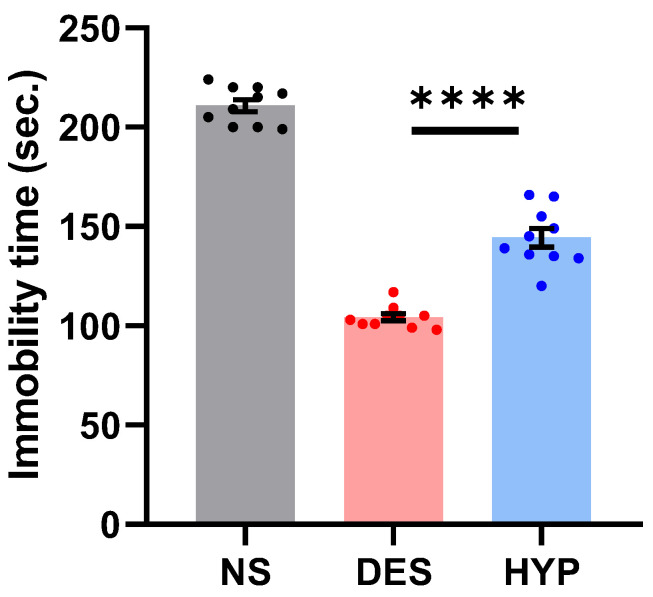
Effect of hypericin (F1) on immobility time in the forced swim test. Statistically significant assessed by an ordinary one-way ANOVA. Graph bars represent means ± SEM. **** *p* < 0.0001. (*n* = 10 for each group). NS: normal saline, DES: desipramine, HYP: hypericin.

**Figure 7 pharmaceuticals-16-00490-f007:**
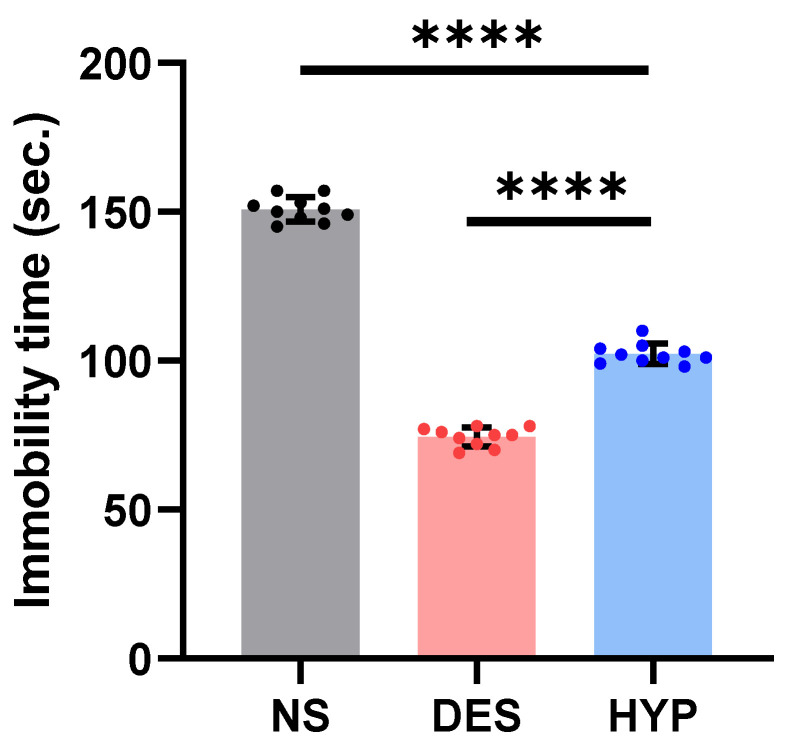
Effect of the administration of hypericin (F1) on immobility time in the tail suspension test. Statistically significant assessed by an ordinary one-way ANOVA. Graph bars represent means ± SEM. **** *p* < 0.0001. (*n* = 10 for each group). NS: normal saline, DES: desipramine, HYP: hypericin.

**Figure 8 pharmaceuticals-16-00490-f008:**
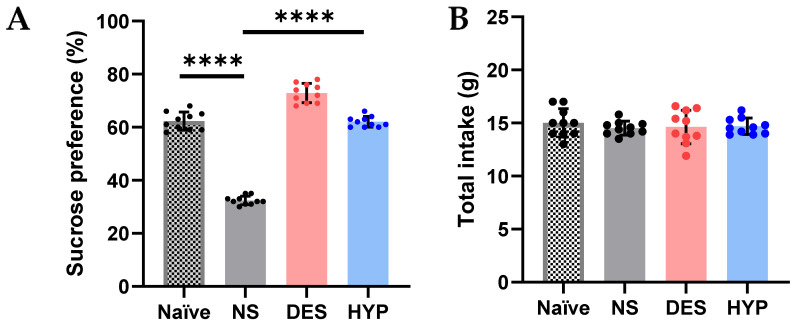
Effect of hypericin (F1) in increasing the preference of sucrose intake in the treated animals, compared to the control group, following chronic mild stress induction (**A**). This increase in preference to sucrose (%) is witnessed in the absence of any effect on total intake (**B**). Statistically significant assessed by an ordinary one-way ANOVA. Graph bars represent means ± SEM. **** *p* < 0.0001. (*n* = 10 for each group). NS: normal saline, DES: desipramine, HYP: hypericin.

**Table 1 pharmaceuticals-16-00490-t001:** Zeta size analysis of the prepared nano-emulsion, describing the nano-sized ranged formulation.

Formulations	Particle Size (nm)	Zeta Potential (mV)
F1	130 ± 06	+72.7
F2	133 ± 04	+58.5
F3	139 ± 07	+56.5
F4	141 ± 05	+89.6

**Table 2 pharmaceuticals-16-00490-t002:** pH, refractive index, and viscosity analysis results of the prepared nanoemulsions.

Formulations	pH	Refractive Index (*n*)	Viscosity (cP)
F1	5.34	1.3425	11.61
F2	6.11	1.3432	15.70
F3	5.90	1.3422	28.09
F4	5.14	1.3491	49.28

**Table 3 pharmaceuticals-16-00490-t003:** Composition of the formulations for the hypericin nanoemulsion.

Formulations	Oil Phase (mL)	Aqueous Phase (mL)	Chitosan (%)
F1	0.5	9.5	0.4
F2	0.5	9.5	0.5
F3	0.5	9.5	0.6
F4	0.5	9.5	0.7

Each formulation contained equal quantities of the drug (2.5 mg), Span^®^ 80, and Tween^®^ 80 (0.5 mL).

## Data Availability

The raw data are available upon reasonable request from the corresponding author.

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
