# Peer review of "Production, Characterization, and In Vitro and In Vivo Studies of Nanoemulsions Containing St. John’s Wort Plant Constituents and Their Potential for the Treatment of Depression"

_pharmaceuticals, 2023, doi:10.3390/ph16040490_

Round 1
Reviewer 1 Report (Previous Reviewer 1)
The manuscript deals with the production and characterization of a nanoemulsion aimed to deliver hypericin. Please, the dispersed phase are not "particles"!
Check English. Typos and grammar mistakes are still in the text.
Both in “Methods” and “Results and Discussion” section avoid unnecessary chapters, e.g. chapter 2.1 and 2.2. are unnecessary; many of them can be grouped with other following subchapters and renamed using a general title, e.g. “Characterization of the systems”
Abstract: it includes only data and results of formulative study. Add comment on potential treatments of depression.
Introduction: I suggest shortening.
Viscosity: add details concerning spindle and rpm.
In “Results and Discussion” avoid repeating methodological details or move in “Methods” description.
“In vivo test”: add which nanoemulsion is used.
From line 199 to 223, both paragraphs are to be rewritten; the content is not clear and style to be improved.
Figure 4: are there any significant differences? Explain which to support the results.
Figure 6, 7, 8: add meaning of abbreviations. Check abbreviations in the text.
Author Response
The manuscript deals with the production and characterization of a nanoemulsion aimed to deliver hypericin. Please, the dispersed phase are not "particles"!
Query: Check English. Typos and grammar mistakes are still in the text.
Reply: Respected reviewer, the manuscript now has been checked by the native English speaker and hope so you will be satisfied with the revision in terms of grammar and typos.
Query: Both in “Methods” and “Results and Discussion” section avoid unnecessary chapters, e.g. chapter 2.1 and 2.2. are unnecessary; many of them can be grouped with other following subchapters and renamed using a general title, e.g. “Characterization of the systems”
Reply: Respected reviewer, as per your kind suggestion we revised the heading and subheadings.
Query: Abstract: it includes only data and results of formulate study. Add comment on potential treatments of depression.
Reply: Respected reviewer, as per your kind suggestion we revised the abstract and added results on depression treatment.
Query: Introduction: I suggest shortening.
Reply: Respected reviewer, as per your kind suggestion we revised the introduction section.
Query: Viscosity: add details concerning spindle and rpm.
Reply: Respected reviewer, as per your kind suggestion we mentioned the viscometer parameters in method section; "spindle no. 2 at 50 rpm".
Query: In “Results and Discussion” avoid repeating methodological details or move in “Methods” description.
Reply: Respected reviewer, as per your kind suggestion we revised both sections.
Query: “In vivo test”: add which nanoemulsion is used.
Reply: Respected reviewer, F1 was used for said analysis
Query: From line 199 to 223, both paragraphs are to be rewritten; the content is not clear and style to be improved.
Reply: Respected reviewer, as per your kind suggestion we revised the said lines 199 to 223.
Query: Figure 4: are there any significant differences? Explain which to support the results.
Reply: Respected reviewer, visually we can say there is no significant difference but inter analysis of release from all formulations has significant difference and results were also supported by other analysis performed.
Query: Figure 6, 7, 8: add meaning of abbreviations. Check abbreviations in the text.
Reply: Respected reviewer, meaning of abbreviations added in the figures captions and also checked in the text.
Reviewer 2 Report (New Reviewer)
In the manuscript titled “Production, Characterization, In-Vitro and In-Vivo Studies of Nanoemulsion containing St. John’s Wort Plant Constituents and its potential for the Treatment of Depression”, the development of nanoemulsion for the treatment of depression was investigated.
1. Extensive editing of the English language and style required for the whole manuscript.
2. The abstract section has a lot of grammatical mistakes and writing errors. This section needs to be rewritten.
3. Same with the Introduction, also it is too lengthy and does not establish the foundation of the study.
4. Scientific name of the plant should be represented as Hypericum perforatum.
5. Line 120, 243 should be mentioned with its unit (nm)?
6. Zeta potential of the prepared nanoemulsion was found to be 58 mV, ideally for a nanoemulsion, the zeta potentials of greater than +30 mV or less than -30 mV are considered stable. How the authors justify that their nanoemulsion is stable?
7. Resolution of the SEM micrograph is poor and also not captured at a suitable magnification. The authors mentioned that the particles were in the range of 130±06 to 141±05 nm, in the SEM micrograph there are particles of 50 nm, 57 nm, 790 nm, and even bigger ones are there. The authors prepared a nanoemulsion, there should be globules, not particles, where are these particles coming from?
8. What is the molecular weight cut off of the dialysis membrane used? Is the drug release study performed in triplicate? Is there any scientific reason for using the USP II apparatus as this study uses a lot of dissolution medium (500 ml) as the same can be achieved using a beaker, and magnetic bead on a stir plate?
9. The conclusions section can be improved supported by the results of the study.
Author Response
In the manuscript titled “Production, Characterization, In-Vitro and In-Vivo Studies of Nanoemulsion containing St. John’s Wort Plant Constituents and its potential for the Treatment of Depression”, the development of nanoemulsion for the treatment of depression was investigated.
Query: Extensive editing of the English language and style required for the whole manuscript.
Reply: Respected reviewer, the manuscript now has been checked by the native English speaking editor and hope so you will be satisfied with the revision in terms of grammar and typos.
Query: The abstract section has a lot of grammatical mistakes and writing errors. This section needs to be rewritten.
Reply: Respected reviewer, the abstract now been revised in revised version of manuscript.
Query: Same with the Introduction, also it is too lengthy and does not establish the foundation of the study.
Reply: Respected reviewer, as per your kind suggestion we revised the introduction section.
Query: Scientific name of the plant should be represented as Hypericum perforatum.
Reply: Respected reviewer, as per your kind suggestion we revised the scientific name.
Query: Line 120, 243 should be mentioned with its unit (nm)?
Reply: Respected reviewer, as per your kind suggestion we mentioned the units in said lines.
Query: Zeta potential of the prepared nanoemulsion was found to be 58 mV, ideally for a nanoemulsion, the zeta potentials of greater than +30 mV or less than -30 mV are considered stable. How the authors justify that their nanoemulsion is stable?
Reply: Respected reviewer, we discussed the results of stability in the discussion section page 3. It has also been reported in literature that emulsion with zeta potential around 50, were found to be of good stability [1].
Query: Resolution of the SEM micrograph is poor and also not captured at a suitable magnification. The authors mentioned that the particles were in the range of 130±06 to 141±05 nm, in the SEM micrograph there are particles of 50 nm, 57 nm, 790 nm, and even bigger ones are there. The authors prepared a nanoemulsion, there should be globules, not particles, where are these particles coming from?
Reply: Respected reviewer, we apologies for the inconvenience caused these are globules of emulsion and mistakenly we wrote as particles. We have re-tested the formulation and new image has been added, that is showing comparatively uniformed size globular structure.
Query: What is the molecular weight cut off of the dialysis membrane used? Is the drug release study performed in triplicate? Is there any scientific reason for using the USP II apparatus as this study uses a lot of dissolution medium (500 ml) as the same can be achieved using a beaker, and magnetic bead on a stir plate?
Reply: Respected reviewer, in the current study we used the dialysis membrane of size 12KD.
As the drug is hydrophobic, so in order to maintain the proper sink condition 500 ml dissolution medium in USP II dissolution apparatus was used. Moreover, in literature various studies has reported the USP II dissolution apparatus for dissolution studies.
Query: The conclusions section can be improved supported by the results of the study.
Reply: Respected reviewer, we revised the conclusion part both in abstract and manuscript.
- Grumezescu, A.M., Nanomaterials for drug delivery and therapy. 2019: William Andrew.
Reviewer 3 Report (New Reviewer)
The main concept of the research is interesting and worth investigating. However, Authors should pay attention to the following aspects of the paper:
1) The novelty of the research should be clearly emphasized in the Abstract.
2) Line 90: there should be a space between the value and the unit ("243 nm" instead of "243nm"). The same applies to other notations (e.g. line 164 etc.).
3) Section 2.1.: according to the definition term "nano" refers to the materials whose at least dimension is within the range 1 - 100 nm. Presented research showed the particle sizes about 130 -140 nm thus the explanation of terming them as "nanoparticles" should be added.
4) The discussion over the results obtained as well as many theoretical information (including e.g. the cationic nature of chitosan) should be supported to a greater extent by adequate literature references.
5) Section 3.1.: the materials applied should be characterized in more detail. For example, there is no information concerning the average molecular weight of chitosan or the chemical composition of eucalyptus oil. This data should be provided.
6) The explanation of the use of the oil phase in the form of eucalyptus oil needs to be given.
7) There is no information concerning the units of the parameters given in the equation visible in line 323.
8) Section 3.3.1.: there is no information concerning the temperature in which the zeta potential measurements were performed.
9) Final conclusions should contain some quantified data.
10) Section References should be prepared in line with the requirements of the Journal and be consistent. Currently, some references contain the whole journal names, and some contain their abbreviations.
Author Response
The main concept of the research is interesting and worth investigating. However, Authors should pay attention to the following aspects of the paper:
Query#1: The novelty of the research should be clearly emphasized in the Abstract.
Query reply: Dear reviewer, a novelty statement has been added in the abstract.
Query#2: Line 90: there should be a space between the value and the unit ("243 nm" instead of "243nm"). The same applies to other notations (e.g. line 164 etc.).
Query reply: The units are corrected as per kind suggestions.
Query#3: Section 2.1.: according to the definition term "nano" refers to the materials whose at least dimension is within the range 1 - 100 nm. Presented research showed the particle sizes about 130 -140 nm thus the explanation of terming them as "nanoparticles" should be added.
Query reply: Yes, you are right, as; a nanoparticle or ultrafine particle is usually defined as a particle of matter that is between 1 and 100 nanometres (nm) in diameter. The term is sometimes are used for larger particles, up to 500 nm. A large number of studies have reported and claimed the nanoparticle in the stated range. [1-2]
1- Jaiswal, M., Dudhe, R., & Sharma, P. K. (2015). Nanoemulsion: an advanced mode of drug delivery system. 3 Biotech, 5, 123-127.
2- Saeed, R. M., Dmour, I., & Taha, M. O. (2020). Stable chitosan-based.
Query#4: The discussion over the results obtained as well as many theoretical information (including e.g. the cationic nature of chitosan) should be supported to a greater extent by adequate literature references.
Query reply: Dear reviewer, thanks for your comment, 20 different studies have been added in the results and discussion section to support the outcomes. However, few more references, especially related to chitosan and its used in stabilizing the emulsion or as permeation enhancer has been added.
Query#5: Section 3.1.: the materials applied should be characterized in more detail. For example, there is no information concerning the average molecular weight of chitosan or the chemical composition of eucalyptus oil. This data should be provided.
Query reply: Required information has been updated in the manuscript.
Query#6: The explanation of the use of the oil phase in the form of eucalyptus oil needs to be given.
Query reply: For the preparation of oil in water emulsion, different oils have been used but the used oil was found to be suitable.
Query#7: There is no information concerning the units of the parameters given in the equation visible in line 323.
Query reply: Dear Reviewer, the relevant information is available in table 2 (Table 2. pH, refractive index, and viscosity analysis results of the prepared nanoemulsions). Units of viscosity has been added as centipoise (cP), while refractive index has no units.
Query#8: Section 3.3.1.: there is no information concerning the temperature in which the zeta potential measurements were performed.
Query reply: Dear Reviewer, the studies have been carried out at room temperature, as mentioned and highlighted red in the manuscript.
Query#9: Final conclusions should contain some quantified data.
Query reply: the final conclusion is corrected as per kind suggestion.
Query#10: Section References should be prepared in line with the requirements of the Journal and be consistent. Currently, some references contain the whole journal names, and some contain their abbreviations.
Query reply: the references are corrected as per kind suggestion.
Round 2
Reviewer 1 Report (Previous Reviewer 1)
Please check throughout the text and remove the word "particle(s)".
Author Response
Reviewer 1
Query: Please check throughout the text and remove the word "particle(s)".
Reply: Respected reviewer, as per your kind suggestion we remove the word "particle(s)".
Reviewer 2 Report (New Reviewer)
Methods and results sections need to be described clearly.
There are a lot of scientific flaws in what the authors stated in the manuscript vs their results. Also, the comments that I gave in the first round of the review were not justified by the authors. There are still a lot of grammatical errors and extensive editing of English is necessary. The SEM micrographs are not aligning with the data that was mentioned in the results. There is no novelty/significance in the content, the quality and scientific soundness is low.
Author Response
Reviewer 2
Query: Methods and results sections need to be described clearly.
There are a lot of scientific flaws in what the authors stated in the manuscript vs their results. Also, the comments that I gave in the first round of the review were not justified by the authors. There are still a lot of grammatical errors and extensive editing of English is necessary. The SEM micrographs are not aligning with the data that was mentioned in the results. There is no novelty/significance in the content, the quality and scientific soundness is low.
Reply: Respected reviewer, the results describing the globular size has been tabulated as an average of the size distribution. When we compared it with the SEM outcomes, they are reflecting the similar findings. There may be confusion, as only few of the entities have been ladled, because it is not possible to label all the globular structure.
All the mentioned and described methodology has been supported by suitable references. It indicates that we have followed the prescribed procedure, set by the researchers.
The prepared formulations are of suitability novelty, as, according to best of our knowledge no formulation has been reported using the composition and ratios of the ingredients, as used in current formulations.
Reviewer 3 Report (New Reviewer)
The paper has been sufficiently improved. Authors paid attention to all mentioned comments and doubts. Thus the paper in its current version may be accepted for publication in the Journal.
Author Response
Responses for Academic Editor Notes:
Question: It is absolutely essential to present the results for all four formulations that you mention in the manuscript, and finally discuss and comment which of the 4 formulations is the preferred one. It should be clear which formulation was employed in the in vivo studies and most importantly why this specific formulation was chosen.
Answer: Dear Editor, in-vitro results of all the formulations have been reported in the manuscript. Section, 2.3-2.6 and 2.8 can be seen. Based on the previous tests performed, F1 Nano-emulsion had been selected for the in-vivo studies, as it had most prominently showed a better drug release profile, the initially, the release rate was slow and the extent was greater at the end of the studies. Similarly, the particle size was smallest amongst all of the four formulations etc. Hence; keeping in mind the satisfactory outcome, F1 was carried out for further analysis.
Question: In the current version of the manuscript it is not clear, for most part of the Results section as well as for all Figures (with the exception of Figure 4), which of the four formulations is presented.
Answer: The results have been tabulated for all the formulation and clearly mentioned in the manuscript, for example Table 1, table 2 etc., describing the outcomes of the four formulations.
To address the concerns of reviewer #2 and provide evidence for the particle sizes in Table 1, high quality SEM images of all formulations must be shown, either in the main manuscript or in the Suppl. Mater. Please also include the zeta potential of each formulation in this Table 1, since you mention it in the title of the Table but you are not presenting them.
Results have been updated in the table for zeta potential.
For SEM images, the early SEM images have been replaced on the suggestion of early reviewers. And the present image is the updated one. For nanoemulsion, most often TEM is recommended, so, for better outcome, TEM may be considered, but unfortunately, we do not have facility at our institution and furthermore, currently we do not have access to this facility anywhere. So, it is requested to considered the represented SEM images. The results have been tabulated for all the formulation and clearly mentioned in the manuscript and supplementary file, for example Figure 1 and Supplementary Figures 3A, 3B and 3C.
Concerning the Supplementary Material files, there are just four figures without any explanation for them in the manuscript (only 2 out of the 4 figures are simply mentioned in the manuscript). Please check papers already published in Pharmaceuticals to correctly prepare this file.
As per your kind suggestion we revised the manuscripts and the supplementary figures file.

This manuscript is a resubmission of an earlier submission. The following is a list of the peer review reports and author responses from that submission.
Round 1
Reviewer 1 Report
The manuscript deals with the production and characterization of a nanoemulsion aimed to deliver hypericin.
The title of the manuscript should highlight that the aim of the manuscript is the preparation of nanoemulsions instead of “Nanostructured Lipid Carriers”. The system was designed for the intranasal route but the effect of hypericin was assessed after the administration by other routes. The composition of the nanoemulsions was modified only for the chitosan percentage. This was not so relevant for the system. The design of the formulations could have been more comprehensive. Maybe the aim of the authors was primarily the in vivo evaluation.
Check English. Throughout the manuscript, Hypericin was misspelled.
Introduction.
It is quite simple and lacking in originality.
Line 28-31. Description of nanoemulsion composition is incomplete. Co-surfactants can work better in combination with a surfactant which is generally used in development of the nanoemulsions. An alone surfactant cannot lower the surface tension of both oil and water interface efficiently to make the nanoemulsions
Line 52. Tween and Span are regularly used, among others, as surfactant.
Line 57. For co-surfactant, see also doi.org/10.1016/j.jddst.2019.03.006.
Line 60. After description of nanoemulsion preparation method, go to line for a new paragraph.
Materials and Methods
In general, many details on equipment used and procedures were not reported.
Nanoemulsion composition. A pseudo-ternary phase diagram was not used to fix the amounts of each component. How were these amounts set? What is the hypercin amount? Was it completely solubilized in the vehicle? In what amount hypericin solution was used?
Line 184. The test was performed using an artificial membrane. This is a “release test”, which considers the ability of the system to release hypericin and cannot provide information about its behaviour when applied on a biological membrane.
Line 206. It is unclear which formulation was used for the in vivo test and the relevance of the administration mode with respect to the intranasal route.
Line 234. What is “Nave” group?
Results and discussion.
In general, in order to see effects on the formulation, results are conditioned by the limited number of formulation and limited parameters considered. Therefore, discussion on the technological characterization is poor.
Line 324. Test was performed with cellophane, an artificial membrane. Discuss results on this starting point. Discuss the increased viscosity with respect to the chitosan amount added. Release profiles are not statistically different. Which formulation was selected?
Author Response
Query: The title of the manuscript should highlight that the aim of the manuscript is the preparation of nanoemulsions instead of “Nanostructured Lipid Carriers”. The system was designed for the intranasal route but the effect of hypericin was assessed after the administration by other routes. The composition of the nanoemulsions was modified only for the chitosan percentage. This was not so relevant for the system. The design of the formulations could have been more comprehensive. Maybe the aim of the authors was primarily the in vivo evaluation.
Query reply: Respected reviewer, as recommended by you we revised the manuscript, now revised file is according to your suggestions.
Query: Check English. Throughout the manuscript, Hypericin was misspelled.
Query reply: Respected reviewer, as recommended by you we correct the spells and also check the English.
Query: It is quite simple and lacking in originality.
Query reply: Respected reviewer, as recommended by you we revised the manuscript and mentions its originality.
Query: Line 28-31. Description of nanoemulsion composition is incomplete. Co-surfactants can work better in combination with a surfactant which is generally used in development of the nanoemulsions. An alone surfactant cannot lower the surface tension of both oil and water interface efficiently to make the nanoemulsions
Query reply: Respected reviewer, as recommended by you we added the description of the components in revised version of manuscript.
Query: Line 52. Tween and Span are regularly used, among others, as surfactant.
Query reply: Respected reviewer, yes both are regularly used in manuscript.
Query: Line 57. For co-surfactant, see also doi.org/10.1016/j.jddst.2019.03.006.
Query reply: Respected reviewer, as recommended by you we added data from suggested article in revised version of manuscript.
Query: Line 60. After description of nanoemulsion preparation method, go to line for a new paragraph.
Query reply: Respected reviewer, as recommended by you changes made in revised version of manuscript.
Query: In general, many details on equipment used and procedures were not reported.
Query reply: Respected reviewer, details of equipment and procedure added for all in revised version of manuscript.
Query: Nanoemulsion composition. A pseudo-ternary phase diagram was not used to fix the amounts of each component. How were these amounts set? What is the hypercin amount? Was it completely solubilized in the vehicle? In what amount hypericin solution was used?
Query reply: Respected reviewer, details were added in revised version of manuscript.
Query: Line 184. The test was performed using an artificial membrane. This is a “release test”, which considers the ability of the system to release hypericin and cannot provide information about its behaviour when applied on a biological membrane.
Query reply: Respected reviewer, we revised the said query in revised version of manuscript.
Query: Line 206. It is unclear which formulation was used for the in vivo test and the relevance of the administration mode with respect to the intranasal route.
Query reply: Respected reviewer, we mentioned in the revised version about used formulation.
Query: Line 234. What is “Nave” group?
Query reply: Respected reviewer, the nave group is negative control group.
Query: In general, in order to see effects on the formulation, results are conditioned by the limited number of formulation and limited parameters considered. Therefore, discussion on the technological characterization is poor.
Query reply: Respected reviewer, we revised the manuscript now revised version is free from technological errors.
Query: Line 324. Test was performed with cellophane, an artificial membrane. Discuss results on this starting point. Discuss the increased viscosity with respect to the chitosan amount added. Release profiles are not statistically different. Which formulation was selected?
Query reply: Respected reviewer, all mentioned were resolved in revised version of manuscript.
Reviewer 2 Report
The authors present an interesting article entitled "Production, Characterization, In-Vitro And In-Vivo Studies of nanostructured Lipid Carriers (NLC) of Sant Wart John Plant Constituents as Brain Targeting via Intranasal Route for Treatment of Depression" Overall it reads well, but needs some editing before publication.
The abstract is clear.
The introduction is clear and concise.
The Materials and Methods section is clear and concise.
The results and discussion is clear and concise.
The conclusion is clear and concise.
Edits:
Please change all instances of "Sant Wart John" to read "St. John's Wort" (including the title and elsewhere).
For chemical formulae, the numbers after the elements should be subscript (e.g., C30H16O8).
2.3 & 2.3.1: please mention the manufacturer, supplier, location, state, country for the UV-vis.
2.3.2: please mention the manufacturer, supplier, location, state, country for the microscope.
2.3.3: please mention the manufacturer, supplier, location, state, country for the zetasizer.
2.3.5: please mention the manufacturer, supplier, location, state, country for the fluorescence microscope.
2.3.7: please mention the manufacturer, supplier, location, state, country for the refractometer.
2.3.8: please mention the manufacturer, supplier, location, state, country for the viscometer.
2.3.9: please mention the manufacturer, supplier, location, state, country for the SEM.
2.3.10: please mention the manufacturer, supplier, location, state, country for the Franz diffusion cell etc.
2.3.11: please mention the manufacturer, supplier, location, state, country for the centrifuge etc.
2.3.12: please mention the manufacturer, supplier, location, state, country for the apparatus.
2.3.12: please mention the ethical approval.
3.7: please include an image to show the dilutability test results.
3.8: please include an image to show the dye solubility test results.
"Data Availability Statement: The data used to support the findings of this study are included in 442
the article." should be changed to read "Data Availability Statement: The raw data are available upon reasonable request to the corresponding author."
Figures:
Figures: 1, 2, 3, 5 should be put in supplementary information / appendix, not the main paper.
After "Figure 1. Prepared hypercine nanoemulsion." include a description of the samples shown.
Figure 4: needs scale bars on the images - and a note in the legend - scale bar represents X micrometers.
For figure 5, please report the numerical values in the text in section 3.4.
Figure 6: needs scale bar on the image - and a note in the legend - scale bar represents X micrometers.
Author Response
Query: Please change all instances of "Sant Wart John" to read "St. John's Wort" (including the title and elsewhere).
Query reply: Respected reviewer, as recommended by you we revised the manuscript, now revised file is according to your suggestions.
Query: For chemical formulae, the numbers after the elements should be subscript (e.g., C30H16O8).
Query reply: Respected reviewer, we mentioned in the revised version about used formulation.
Query: 2.3 & 2.3.1: please mention the manufacturer, supplier, location, state, country for the UV-vis.
Query reply: Respected reviewer, details of UV-vis equipment and procedure added for all in revised version of manuscript.
Query: 2.3.2: please mention the manufacturer, supplier, location, state, country for the microscope.
Query reply: Respected reviewer, details of microscope equipment and procedure added for all in revised version of manuscript.
Query: 2.3.3: please mention the manufacturer, supplier, location, state, country for the zetasizer.
Query reply: Respected reviewer, details of zetasizer equipment and procedure added for all in revised version of manuscript.
Query: 2.3.5: please mention the manufacturer, supplier, location, state, country for the fluorescence microscope.
Query reply: Respected reviewer, details of fluorescence microscope equipment and procedure added for all in revised version of manuscript.
Query: 2.3.7: please mention the manufacturer, supplier, location, state, country for the refractometer.
Query reply: Respected reviewer, details of refractometer equipment and procedure added for all in revised version of manuscript.
Query: 2.3.8: please mention the manufacturer, supplier, location, state, country for the viscometer.
Query reply: Respected reviewer, details of viscometer equipment and procedure added for all in revised version of manuscript.
Query: 2.3.9: please mention the manufacturer, supplier, location, state, country for the SEM.
Query reply: Respected reviewer, details of SEM equipment and procedure added for all in revised version of manuscript.
Query: 2.3.10: please mention the manufacturer, supplier, location, state, country for the Franz diffusion cell etc.
Query reply: Respected reviewer, details of Franz diffusion cell equipment and procedure added for all in revised version of manuscript.
Query: 2.3.11: please mention the manufacturer, supplier, location, state, country for the centrifuge etc.
Query reply: Respected reviewer, details of centrifuge equipment and procedure added for all in revised version of manuscript.
Query: 2.3.12: please mention the manufacturer, supplier, location, state, country for the apparatus.
Query reply: Respected reviewer, details of apparatus equipment and procedure added for all in revised version of manuscript.
Query: 2.3.12: please mention the ethical approval.
Query reply: Respected reviewer, ethical approval statement is mentioned in the In-Vivo method
Query: 3.7: please include an image to show the dilutability test results.
Query reply: Respected reviewer, the image of dilutability is included as shown in Figure 5
Query: 3.8: please include an image to show the dye solubility test results.
Query reply: Respected reviewer, the image of dye solubility test results is included as shown in Figure 2.
Query: "Data Availability Statement: The data used to support the findings of this study are included in the article." should be changed to read "Data Availability Statement: The raw data are available upon reasonable request to the corresponding author."
Query reply: Respected reviewer, as recommended by you we revised the "Data Availability Statement: The raw data are available upon reasonable request to the corresponding author.", now revised file is according to your suggestions.
Query: Figures: 1, 2, 3, 5 should be put in supplementary information / appendix, not the main paper.
Query reply: Respected reviewer, Figures 1, 2, 3, 5 were removed from the main paper and put them in supplementary information.
Query: After "Figure 1. Prepared hypercine nanoemulsion." include a description of the samples shown.
Query reply: Respected reviewer, the description of prepared hypericin were included in the related section.
Query: Figure 4: needs scale bars on the images - and a note in the legend - scale bar represents X micrometers.
Query reply: Respected reviewer, scale bars of Figures 4 was added on the image, and a note in the legend - scale bar represents as X micrometers.
Query: For figure 5, please report the numerical values in the text in section 3.4.
Query reply: Respected reviewer, the numerical values of Figure 5 were included in the text in the related section.
Query: Figure 6: needs scale bars on the images - and a note in the legend - scale bar represents X micrometers.
Query reply: Respected reviewer, scale bars of Figures 6 was added on the image, and a note in the legend - scale bar represents as X micrometers.
Round 2
Reviewer 1 Report
A very limited number of suggestions proposed in my first Report have been taken into account in the version of the manuscript available here. Could it have been a mistake in the upload process?
Author Response
Reviewer 1
Query: The title of the manuscript should highlight that the aim of the manuscript is the preparation of nanoemulsions instead of “Nanostructured Lipid Carriers”. The system was designed for the intranasal route but the effect of hypericin was assessed after the administration by other routes. The composition of the nanoemulsions was modified only for the chitosan percentage. This was not so relevant for the system. The design of the formulations could have been more comprehensive. Maybe the aim of the authors was primarily the in vivo evaluation.
Query reply: Respected reviewer, as recommended by you we revised the title of manuscript, and now in revised file line 1-4 represents modified title with yellow highlights. Our research group has been working on different aspects of the drug, its different formulations/dosage forms as well as routes of drug administration. Unfortunately, some of the data/literature regarding the intranasal route has been added in this manuscript, which was not the desired one for this manuscript. The said section has been removed form the text and manuscript has been updated accordingly. Different concentrations of oil/surfactant/co-surfactants have been used in the piolet studies. Different stabilizer/thickening agent etc. has all been evaluated. Finally, the used concentrations of the chitosan were found suitable for stable emulsion. That’s why the formulation having varying concentration of the chitosan have been reported here for the readers.
Query: Check English. Throughout the manuscript, Hypericin was misspelled.
Query reply: Respected reviewer, the manuscript was rechecked by an English native speaker and also hypericin spell were corrected in revised version of manuscript.
Query: It is quite simple and lacking in originality.
Query reply: Respected reviewer, very few studies were performed that represent the potential of St. John’s Wort Plant Constituents as a treatment option for depression, we also tried to add more value in revised version of manuscript. Moreover, the preparation of nanoemulsion having the composition, as did in the current studies was novel, as per best of our knowledge.
Query: Line 28-31. Description of nanoemulsion composition is incomplete. Co-surfactants can work better in combination with a surfactant which is generally used in development of the nanoemulsions. An alone surfactant cannot lower the surface tension of both oil and water interface efficiently to make the nanoemulsions
Query reply: Respected reviewer, the description for nanoemulsion constituents were added in line 24-42 in revised version and also highlighted with yellow marker.
Query: Line 52. Tween and Span are regularly used, among others, as surfactant.
Query reply: Respected reviewer, the literature regarding the use of tween and span was added in revised version in line 52-54 and 56-59. Moreover, we have tried to use the commonly available ingredients to make it cost effective formulation as well.
Query: Line 57. For co-surfactant, see also doi.org/10.1016/j.jddst.2019.03.006.
Query reply: Respected reviewer, the literature suggested by you was added in revised version of manuscript. (Reference#16)
Query: Line 60. After description of nanoemulsion preparation method, go to line for a new paragraph.
Query reply: Respected reviewer, as recommended by you changes made in revised version of manuscript.
Query: In general, many details on equipment used and procedures were not reported.
Query reply: Respected reviewer, details of each equipment, its use, procedure and company data was added in revised version of manuscript.
Query: Nanoemulsion composition. A pseudo-ternary phase diagram was not used to fix the amounts of each component. How were these amounts set? What is the hypercin amount? Was it completely solubilized in the vehicle? In what amount hypericin solution was used?
Query reply: Respected reviewer, we agreed with your deep insight about pseudo-ternary phase diagram and tried to mentioned all amounts for each component along with hypercin amount. Yes, it was completely solubilized in the vehicle and amount was mentioned in method section in revised version of manuscript.
Query: Line 184. The test was performed using an artificial membrane. This is a “release test”, which considers the ability of the system to release hypericin and cannot provide information about its behaviour when applied on a biological membrane.
Query reply: Respected reviewer, the appraisal of nanoemulsion was checked using in-vitro, & in-vivo. The formulation, showing better outcomes were evaluated for in-vivo studies, and the findings have been described well in the manuscript. Furthermore, we will keep your valuable suggestions for the next project, as we are working different formulation and route of administration for the said drug.
Query: Line 206. It is unclear which formulation was used for the in vivo test and the relevance of the administration mode with respect to the intranasal route.
Query reply: Respected reviewer, the formulation F2 was used in in-vivo test as it has smallest particle size, good zeta potential, good in-vitro permeation, and stable formulation like others.
Moreover, as stated earlier, that we are working on different aspects, different formulations well as different routes of administration. Here, the manuscript has been revised and updated with oral delivery of the prepared formulation.
Query: Line 234. What is “Nave” group?
Query reply: Respected reviewer, the nave group is negative control group.
Query: In general, in order to see effects on the formulation, results are conditioned by the limited number of formulation and limited parameters considered. Therefore, discussion on the technological characterization is poor.
Query reply: Respected reviewer, we tried to add more discussion on the said test and revised version is much better form of manuscript, we also tried to remove technological issues.
Query: Line 324. Test was performed with cellophane, an artificial membrane. Discuss results on this starting point. Discuss the increased viscosity with respect to the chitosan amount added. Release profiles are not statistically different. Which formulation was selected?
Query reply: The said section has been improved as per kind suggestion. Discussion related to viscosity has been added. In the relevant section. Respected reviewer, we have prepared a lot number of formulations with different types of excipients along with surfactants, oil etc., such as beta-cyclodextrin, polyvinyl pyrolidone and chitosan etc. There were different numbers of trial and out of those we selected 4 formulations with varying concentration of chitosan. These formulations, and amongst them, F2 showed better outcomes, which has been processed for in-vivo studies.
Round 3
Reviewer 1 Report
There are still typos and grammatical mistakes in the text.
Line 195, 196. Line 389: the authors' comments in 2.9 In-Vitro Permeation Studies are questionable. Released profiles in Figure 4 are not statistically different. Is it possible that drug released reached 100%? What is solubility in the receptor phase?
Permeation studies were performed using biological membrane. The experiment in the manuscript was performed using an artificial membrane. This is a “release test”, which considers the ability of the system to release hypericin and cannot provide information about its behaviour when applied on a biological membrane.
Author Response
I would like to thank reviewer for his valuable comments. We carefully revised our manuscript according to the referees' comments.
Answers to the reviewer’s comments:
Query: There are still typos and grammatical mistakes in the text.
Answer: The article has been revised comprehensively for grammatical mistakes
Query: Line 195, 196. Line 389: the authors' comments in 2.9 In-Vitro Permeation Studies are questionable. Released profiles in Figure 4 are not statistically different. Is it possible that drug released reached 100%? What is solubility in the receptor phase?
Answer: Dear Reviewer, thanks for your comments, it is tried to improve the text in stated sections (line 195, 196 and 389).
All the formulations have not shown the release exactly 100%. Furthermore, it is comprehensively available in the literature, that by reducing the particle size of the drug, solubility can be improved. In the current studies, author has tried to prepare the nano-emulsion, having the size range less than 150nm, hence; the question of solubility can be resolved here. It is well established concept that by decreasing the particles size, one can increase the surface area of the drug, leading to increase in the rate of hydration and hence the solubility. Furthermore, the formulations have hydrophilic as well as hydrophobic surfactants, and both of these are known to have the potential to improve the solubility as well as the ability to improve the permeation of the drugs. [1-4].
As, here in the studies, the objective was not the improvement or solubility enhancement, so this was the reason that both of the stated objectives were not addressed comprehensively. However, we have taken this comments and will consider in our future studies.
Query: Permeation studies were performed using biological membrane. The experiment in the manuscript was performed using an artificial membrane. This is a “release test”, which considers the ability of the system to release hypericin and cannot provide information about its behaviour when applied on a biological membrane.
Answer: In the literature, a large number of researchers have reported the permeation studies across the artificial membrane, as we did in current studies. Different types of membranes have been used for such studies. We do appreciate your positive comments and recommendation, and surely will be considered in our future studies [5-8].
- Hussain, K., et al., Impact of particle-size reduction on the solubility and antidiabetic activity of extracts of leaves of Vinca rosea. 2019. 16(3): p. 335.
- Brayden, D.J. and S.J.E.O.o.D.D. Maher, Transient Permeation Enhancer®(TPE®) technology for oral delivery of octreotide: a technological evaluation. 2021. 18(10): p. 1501-1512.
- Hussain, A., et al., Optimized permeation enhancer for topical delivery of 5-fluorouracil-loaded elastic liposome using Design Expert: part II. 2016. 23(4): p. 1242-1253.
- El-Nabarawi, M.A., et al., In vitro skin permeation and biological evaluation of lornoxicam monolithic transdermal patches. 2013. 5(2): p. 242-248.
- Kollipara, R.K., et al., Curcumin loaded ethosomal vesicular drug delivery system for the treatment of melanoma skin cancer. 2019. 12(4): p. 1783-1792.
- Kumar, S., et al., Comparison Effect of Penetration Enhancer on Drug Delivery System. 2021.
- SETHI, B. and R.J.I.J.A.P. MAZUMDER, COMPARISON OF EFFECT OF PENETRATION ENHANCER ON DIFFERENT POLYMERS FOR DRUG DELIVERY. 2019. 11(1): p. 89-93.
- Kumari, R., et al., EFFECT OF TERPENES AS PENETRATION ENHANCERS ON THE RELEASE AND PERMEATION KINETICS OF MELOXICAM GELS FORMULATIONS. 2019: p. 1-5.
